# Evaluating leishmanicidal effects of *Lucilia sericata* products in combination with *Apis mellifera* honey using an in vitro model

Jila Sherafati[1,2], Mohammad Saaid Dayer[1]*, Fatemeh Ghaffarifar[1], Kamran Akbarzadeh[3], Majid Pirestani[1]

**1** Faculty of Medical Sciences, Department of Parasitology and Medical Entomology, Tarbiat Modares University, Tehran, Iran, **2** Faculty of Medical Sciences, Student Research Committee, Tarbiat Modares University, Tehran, Iran, **3** Department of Medical Entomology and Vector Control, School of Public Health, Tehran University of Medical Sciences, Tehran, Iran

* dayer@modares.ac.ir

**Data Availability Statement:** All relevant data are within the manuscript and its Supporting Information file.

## Abstract

Leishmaniasis is a zoonotic disease caused by an intracellular parasite from the genus *Leishmania*. Lack of safe and effective drugs has increasingly promoted researches into new drugs of natural origin to cure the disease. The study, therefore, aimed to investigate the anti-leishmanial effects of *Lucilia sericata* larval excretion/secretion (ES) in combination with *Apis mellifera* honey as a synergist on *Leishmania major* using an in vitro model. Various concentrations of honey and larval ES fractions were tested against promastigotes and intracellular amastigotes of *L. major* using macrophage J774A.1 cell line. The inhibitory effects and cytotoxicity of ES plus honey were evaluated using direct counting method and MTT assay. To assess the effects of larval ES plus honey on the amastigote form, the rate of macrophage infection and the number of amastigotes per infected macrophage cell were estimated. The 50% inhibitory concentration (IC$_{50}$) values were 21.66 μg/ml, 43.25 60 μg/ml, 52.58 μg/ml, and 70.38 μg/ml for crude ES plus honey, ES >10 kDa plus honey, ES <10 kDa plus honey, and honey alone, respectively. The IC$_{50}$ for positive control (glucantime) was 27.03 μg/ml. There was a significant difference between viability percentages of promastigotes exposed to different doses of applied treatments compared to the negative control (p ≤ 0.0001). Microscopic examination of amastigote forms revealed that dosages applied at 150 to 300 μg/ml significantly reduced the rate of macrophage infection and the number of amastigotes per infected macrophage cell. Different doses of larval products plus honey did not show a significant toxic effect agaist macrophage J774 cells. The larval ES fractions of *L. sericata* in combination with *A. mellifera* honey acted synergistically against *L. major*.

## Introduction

Leishmaniasis is a complex of vector-borne diseases caused by different species of an obligate intracellular protozoa of the genus *Leishmania* [1, 2]. These zoonotic diseases manifest in three

**Funding:** This study was financially supported by the Student Research Committee, the Faculty of Medical Sciences, Tarbiat Modares University, Tehran, Iran. (Grant no: Med/8288)". We also declare. The funders had no role in study design, data collection and analysis, decision to publish, or preparation of the manuscript.

**Competing interests:** The authors have declared that no competing interests exist.

**Abbreviations:** CL, Cutaneous leishmaniasis; ES, Excretion and secretion; %I, Infection percentage; %DI, Decreased infection percentage; SVI, Survival index; PL, Parasite load. PBS: Phosphate buffered saline; BSA, Bovine serum albumin; NNN, Novy-Macneal-Nicolle; FBS, Fetal bovine serum; RPMI, Roswell Park Memorial Institute; DMEM, Dulbecco's Modified Eagle Medium; MTT, 3-(4.5-Dimethylthiazol-2-yl)-2,5-diphenyltetrazolium bromide; DMSO, Dimethyl sulfoxide; IC50, Inhibitory concentration for 50% of parasite; SDS, PAGE: Sodium dodecyl sulfate-polyacrylamide gel electrophoresis; AUC, Area Under Curve; MMPs, Matrix metalloproteinases.

main forms including cutaneous leishmaniasis (CL), visceral leishmaniasis (VL), and mucocutaneous leishmaniasis (MCL). Species such as *Leishmania major*, *L. tropica*, *L. infantum* and *L. aethopica* in the Old World and *L. perivuana*, *L. braziliensis* and *L. mexicana* in the New World are transmitted by infected bites of different sand-fly species to cause CL in humans [3]. The increasing prevalence of CL in new geographical areas has been attributed to poverty, immune-related diseases, migration, disasters, and drug resistance [4, 5]. Despite application of many control and prevention programs, CL is still persistent in some areas due to widespread insect vectors, poor health, income level, and drug resistance (CDC ref). About 95% of CL cases occur in the Americas, the Mediterranean basin, the Middle East, and Central Asia. The new CL cases are estimated at between 600,000 and 1 million worldwide annually [6].

CL is more likely to cause dry (urban) and wet (rural) wounds leaving ugly scars in exposed areas of the body such as the hands and the face, imposing economic and health burdens [7, 8]. The routine therapies of CL include using pentavalent antimony compounds (Sb), thermotherapy (hot and/or cryo), and topical and intralesional administration of Paromomycin, Amphotericin B and Pentamidine. The former being the first line therapy in many countries. However, the antimonial drugs are associated with clinical side effects such as cardiac toxicity, clinical pancreatitis, arthralgia, myalgia, nausea/vomiting, liver transaminase abnormalities, pancytopenia, and renal toxicity. This is why many studies are underway to find alternative natural compounds [9–11]. Nowadays, researchers have presented evidence that natural compounds including insect products could be promising candidates for development of new anti-*Leishmania* agents [12, 13].

The larvae of *Lucilia sericata* (Diptera: Calliphoridae) are used for maggot therapy of drug-resistant chronic wounds [14]. The maggot therapy results in removal of necrotic tissue, stimulation of granulation and production of antiseptics against pathogenic microorganisms such as gram-positive and negative bacteria, fungi, and parasites, particularly *Leishmania* [15–17]. In addition, larval excretion/secretion (ES) of *L. sericata* involves of proteolytic enzymes (trypsin-like and chymotrypsin-like serine proteases, metalloproteinase and aspartyl proteinases) which are able to digest bacteria and necrotic tissue [18, 19].

On the other hand, honey produced by *Apis mellifera* (Hymenoptera: Apoidea) has strong antibacterial activity and helps clearing the infection [20, 21]. Clinical and laboratory evidences showed that honey, also, has antimicrobial properties against a wide range of viruses, fungi and protozoa [22–24]. The antibacterial activity of honey has been attributed to its natural hydrogen peroxide content [25]. The production of hydrogen peroxide by glucose oxidase in honey does not cause any tissue damage but prevents microbial invasion [25–27]. Therefore, honey has been effective in healing CL lesions [28].

In the present study, anti-leishmanial activity of honey in combination with crude and fractionated ES of *L. sericata* larvae was investigated in an in vitro model.

## Materials and methods

### Ethics statement

This study did not involve animals and was performed in vitro. However, the Ethics committee of Tarbiat Modares University approved all protocols described in the current study (Approval No. IR.MODARES.REC.1399.124).

### Insect rearing and larval ES collection

To obtain larval ES, about 100 stage II and III larvae of *L. sericata* were selected from previously established colonies and disinfected as described by Cruz-Saavedra et al [29, 30]. The ES

was collected by centrifugation at 4000 g for 10 minutes [17]. Finally, the upper phase was removed and used for experimentation.

## Fractionation of the crude ES

The fractionation of the crude ES into two fractions of >10 and <10 kDa was performed through amicon ultra_4 fillers by centrifugation at 7500 g for 40 minutes following the manufacturer's instructions. For sterilization, both the crude ES and its fractions were passed through 0.22 μm syringe filter. The sterilized products were cultured on agar blood medium and incubated at 35°C for 48 h to ensure no microbial growth happens. The crude ES and its fractions were then stored at -20°C until use.

## Bradford protein assay

The amount of protein in solution was determined using a standard curve obtained by ploting OD readings against a serial dilution of Bovine serum albumin (BSA) concentrations (0, 31.25, 62.5, 125, 250, 500, 1000 μg/ml) tittered by the macro assay (No.DB9684-50ml) kit, following the manufacturer's instructions. The reactions took palce in 8-well plates upon incubation at 25°C for 5 minutes. The asorbance was measured using an ELISA reader (Model 680, BIORAD) at 570 nm [31].

## Analysis of larval ES products by SDS-PAGE

The protein profile of the crude and fractionated larval ES (>10 and <10 kDa) of *L. sericata* were analyzed using 1-mm-thick 12.5% Sodium dodecyl sulfate–polyacrylamide gel at a 110-V constant voltage using "Mini-Protein III" system (Bio-Rad) under reduction conditions. The gels were then stained with a solution containing Coomassie brilliant blue G250 and methanol. The molecular weights of the protein bands were estimated in comparison with a prestained protein ladder (PAGEmark, 786–418). About 20–25 protein bands were observed for crude ES, which corresponded to bands in both the >10 kDa and <10 kDa fractions. The protein profiles of larval ES and its fractions were presented in our previous study [30]. The profile obtained from ES separation showed that the separation was done with high accuracy.

## Honey preparation as a synergist

The synergistic effects of honey in combination with either the crude ES or ES fractions were tested using honey purchased from a beekeeper (bees fed on *Astragalus* spp) from Saqqez City in Kurdistan Province. The honey was purely natural as no sucrose was used to feed the bees. The honey solution was first filter sterilized and then tested for any microbial contamination on blood agar medium by incubation at 35°C [32]. In the laboratory, honey solutions were prepared by dilution with distilled water to obtain the following concentrations: 350, 300, 250, 200, 150, 100, 50 and 25 μg / ml [33, 34]. Finally, the solutions were stored at -20°C until used [33].

## Parasite cultivation

*L. major* strain MRHO/IR/75/ER was obtained from Pasteur Institute of Iran. The promastigotes were incubated in Roswell Park Memorial Institute (RPMI) 1640 medium (Gibco, USA) supplemented with 10% heat-inactivated fetal bovine serum (FBS) (Gibco, USA) and 100 μg/mL penicillin-streptomycin (Thermo Fisher Scientific, USA) at 26°C.

## Culture of the murine macrophage cell line

The mouse macrophage cell line J774A.1 was purchased from Pasteur Institute of Iran (https://fa.pasteur.ac.ir/) registered under the Cell Bank Code: C483. The cell line J774A.1 was cultured in Dulbecco's Modified Eagle Medium (DMEM) (Gibco, USA) supplemented with 10% heat-inactivated FBS (Gibco, USA) and 100 μg/mL penicillin-streptomycin (Thermo Fisher Scientific, USA). The cell suspension was incubated in 25 ml cell culture flasks (Jet Biofil) at 37˚C under a humidified atmosphere of 5% $CO_2$.

## Promastigote assay

Promastigotes of *L. major* were cultured in RPMI medium supplemented with 20% FBS in 96-well plates (SPL, Korea) at a concentration of $1 \times 10^6$ cells/mL in the presence of honey, the crude ES + honey, ES >10 kDa + honey, ES<10 kDa + honey and incubated at 26˚C. A serial dilution was performed from the crude ES and its products plus honey with initial concentration of 350 μg/ml. Glucantime was added to the promastigote culture in the positive control, whereas in the negative control, the cell culture was left without treatment. All experiments were performed in triplicate. In order to determine the parasite viability, the proliferation of promastigotes was determined by cell counting using a hemocytometer (Neubauer chamber) after 24, 48 and 72-h incubation [35].

## Macrophage cytotoxicity test

Mice macrophage cell monolayers (cell line J774A.1) were washed with phosphate buffered saline (PBS) for 5 minutes at 37˚C followed by washing with DMEM medium containing 10% FBS, then centrifuged at 200 g for 10 minutes at 4˚C. The cell suspension was cultured in a 96-well plate (SPL, Korea) at $1 \times 10^6$ cells/mL concentration for 24 h. The cultured cells were then exposed to different concentrations of compounds including honey, honey + crude ES, honey + ES >10 kDa, honey + ES <10 kDa, glucantime (positive control) and PBS (negative control). Then, the plates were incubated at 37˚C for 24 h under 5% $CO_2$ atmosphere. Then, 20 μl of MTT (3-(4.5-dimethylthiazol-2-yl)-2,5-diphenyltetrazolium bromide) solution at final concentration of 5 mg/ml was added to each well. The experimental plates were again incubated at 37˚C for 4 h, followed by removal of the top liquid from the wells. Subsequently, 100 μl of dimethyl sulfoxide (DMSO) was added to each well. The optical absorbance was read after 15 minutes using an ELISA reader (Model 680, BIORAD Co.) at a wavelength of 570 nm [36].

## Amastigotes assay

Macrophage cells (J774A.1) were seeded at $2 \times 10^6$ cells/mL on glass coverslips in 12-well culture plates (SPL, Korea) and incubated at 37˚C under 5% $CO_2$ atmosphere for 24 h [6]. Non-adherent cells were washed with PBS. Adhered macrophages were exposed to $1 \times 10^5$ stationary phase *Leishmania* promastigotes/well and incubated at 37˚C under 5% $CO_2$ atmosphere for 24 h. Extracellular promastigotes in the wells were removed by washing with PBS. Subsequently, test compounds were added to the experimental plates at different concentrations based on $IC_{50}$ values obtained in promastigote assay. The compounds included 1) honey (350 and 150 μg/ml); 2) honey + crude ES (350 and 150 μg/ml); 3) honey + ES >10 kDa (350 and 150 μg/ml); 4) honey + ES <10 kDa (350 and 150 μg/ml); and 5) glucantime (100 and 50 μg/ml). The experiments were done in triplicate. The negative control did not receive any treatment. After 72 hours, the glass coverslips were fixed with methanol, stained with 10% Giemsa and examined by light microscopy using immersion oil. The number of infected macrophages

and the average number of parasites per macrophage in 100 cells were counted randomly in ten fields under a light microscope.

### Statistical analyses

The standard curve and its equation were obtained through online software at (https://www.aatbio.com)) and the graph was replotted using GraphPad Prism in the Saturation Binding Data Graph section. The statistical analyses of results were done using GraphPad Prism version 6.07 by one-sample t-test and ANOVA. Data were presented as the means ± standard deviation (SD). The independence of the two categorical variables was determined using chi-square and / or Fisher's exact tests. The dose-response curve was plotted to determine the $IC_{50}$ of promastigote assay using nonlinear regression. Amastigote infection data was evaluated using the equations presented in [37]. Statistical significance was assigned at the level of ($P<0.05$).

## Results

### Bradford protein assay

Bradford protein assay was performed to calculate the protein concentration of *L. sericata* larval products. The average net absorbance at 595 nm for seven standard dilutions from low to high concentrated were as follows; 0.72, 0.785, 0.825, 0.901, 1.01, 1.25, and 1.84. Average net absorbance at 595 nm for the crude ES, ES >10 kDa, and ES <10 kDa were 1.62, 1.48 and 1.16 respectively. The average protein concentrations for the crude ES, ES >10 kDa, and ES <10 kDa were 809.8, 680.6, and 385.3 μg/ml respectively (Fig 1). Consequently, the maximum concentration used in the current study was 350 μg/ml.

### Synergistic effects of ES products plus honey on promastigotes growth

The $IC_{50}$ values and Area Under the Curve (AUC) of the crude ES + honey, ES >10 kDa + honey, ES <10 kDa + honey, honey, and glucantime were evaluated against promastigotes at

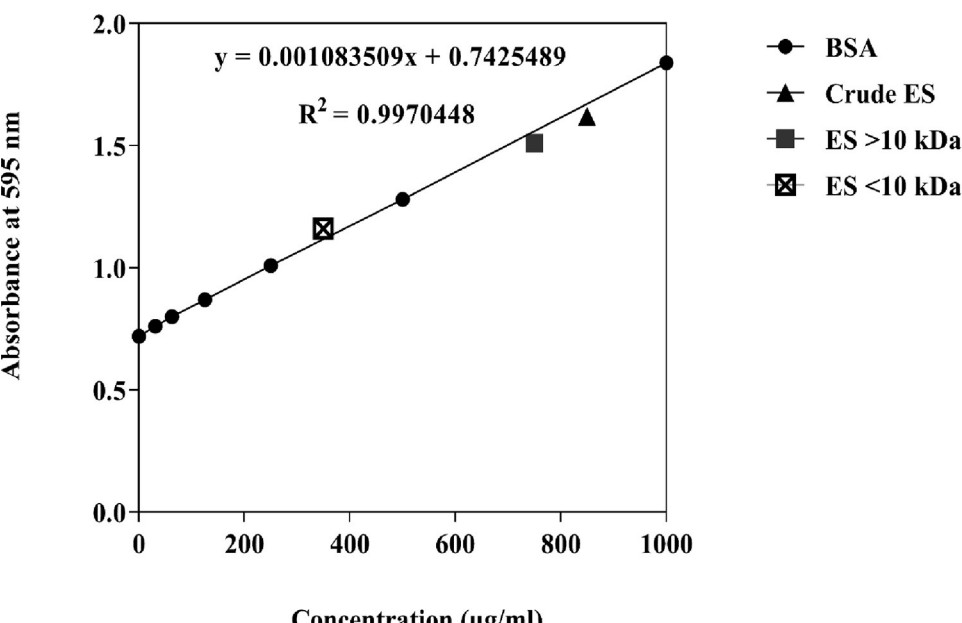

**Fig 1. Bovine serum albumin (BSA) standard curve and OD's obtained for crude and fractionated larval extracts.**

24, 48, and 72 h. The lowest $IC_{50}$ values for the crude ES + honey, ES >10 kDa + honey, ES <10 kDa + honey, honey, and glucantime after 72 h exposure time were 21.66 µg/ml (log = 1.336), 43.25 µg/ml (log = 1.636), 52.58 µg/ml (log = 1.721), 70.38 µg/ml (log = 1.847), and 27.03 µg/ml (log = 1.423) respectively (Fig 2). Regardless of the applied doses, the viability percentages of promastigotes in treated groups was significantly lower than that in the negative control (p ≤ 0.0001). On the other hand, there were no statistically significant differences in the viability percentages of promastigotes treated with the crude ES + honey, ES >10 kDa + honey, and ES <10 kDa + honey compared to glucantime with *P* values equal to 0.4029, 0.9011, and 0.6972 respectively, though the difference was significant between honey and glucantime (*P* = 0.0118). The crude ES + honey showed higher toxicity at low concentrations compared with glucantime. Fig 3 shows that longer exposure time and higher concentration of ES plus honey lead to greater toxicity to promastigotes (Figs 2 and 3).

## Synergistic cytotoxicity of ES products plus honey to macrophages by MTT

MTT test showed that ES products plus honey exerted synergistic effects on the macrophage cell line J774A.1 after 48 and 72 hours in a dose-dependent manner. The crude ES + honey, ES >10 kDa + honey, ES <10 kDa + honey induced the same effects on the viability of macrophage cells as glucantime did with no significant difference between them (*P* > 0/05). The difference between honey and glucantime treatment was significant, in other words, honey showed less toxicity (*P* = 0.0141). However, all treatment groups as well as glucantime were significantly different from the negative control (*P* < 0/0001) (Fig 4).

## Susceptibility of amastigotes to ES products plus honey as a synergist

After 72 hours' exposure time, honey alone and honey plus larval ES products significantly reduced the infection rate of **amastigotes** and inhibited their growth compared to the control group (*P* = 0.004 and *P* = 0.0002, respectively). There was a statistically significant difference between ES <10 kDa plus honey compared to glucantime against amastigotes in terms of infection rate (*P* = 0.048 and *P* = 0.039 respectively) and percentage of growth inhibition (*P* = 0.033 and *P* = 0.004 respectively). There was no significant difference between infection rates of amastigotes exposed to the crude ES plus honey and ES >10 kDa plus honey compared with those exposed to glucantime (*P* = 0.559 and *P* = 0.184). The former treatments inhibited amastigotes growth rates in macrophages in the same manner compared with glucantime at *P* = 0.221 and *P* = 0.453 respectively. The effect of ES <10 kDa plus honey was significantly less that glucantime in term of reduction of infection rate and growth inhibition (*P* = 0.041 and *P* = 0.003 respectively) (Fig 5 and Table 1).

For the negative control group, the parasite load was 3.894 ± 0.091 amastigotes per macrophage, whereas those for groups treated with the crude ES plus honey and ES >10 kDa plus honey at 300 µg/ml concentration were 1.081 ± 0.155 and 1.535 ± 0.046 amastigotes/macrophage cells respectively. The latter treatments both resulted in lower amastigotes/macrophage compared with glucantime applied at 100 µg/ml concentration (1.68 ± 0.061 amastigotes/macrophage). However, treatment with ES <10 kDa plus honey and honey at 300 µg/ml resulted in 2.1 ± 0.014 and 2.273 ± 0.61 amastigotes/macrophage respectively. These were less effective in comparison with glucantime. The survival indexes of amastigotes in groups treated with glucantime and the negative control were 51.927 ± 3.56 and 273.97 ± 1.151 respectively. The survival indexes in groups treated with the crude ES plus honey, ES >10 kDa plus honey, ES <10 kDa plus honey, and honey at 300 µg/ml concentrations were 22.09 ± 1.66, 66.908 ± 8.11, 103.39 ± 3.12, and 130.38 ± 3.994 respectively (Table 1).

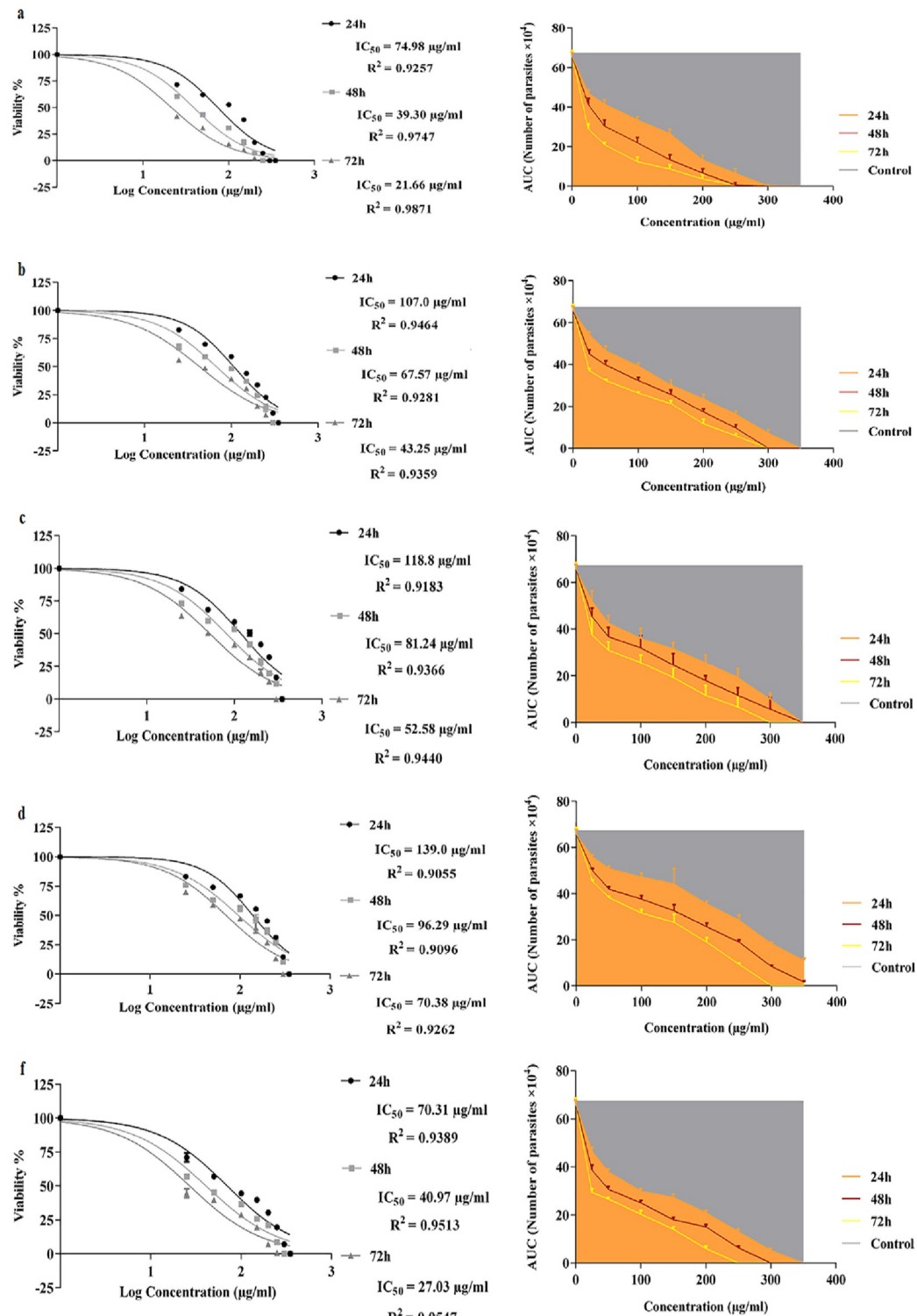

**Fig 2. Dose-response curves and Areas Under Curves related to ES products with honey as a synergist on promastigotes.** (a) crude ES + honey. (b) ES >10 kDa + honey. (c) ES <10 kDa + honey. (d) Honey. (f) Glucantime at 24, 48 and 72 h. intervals.

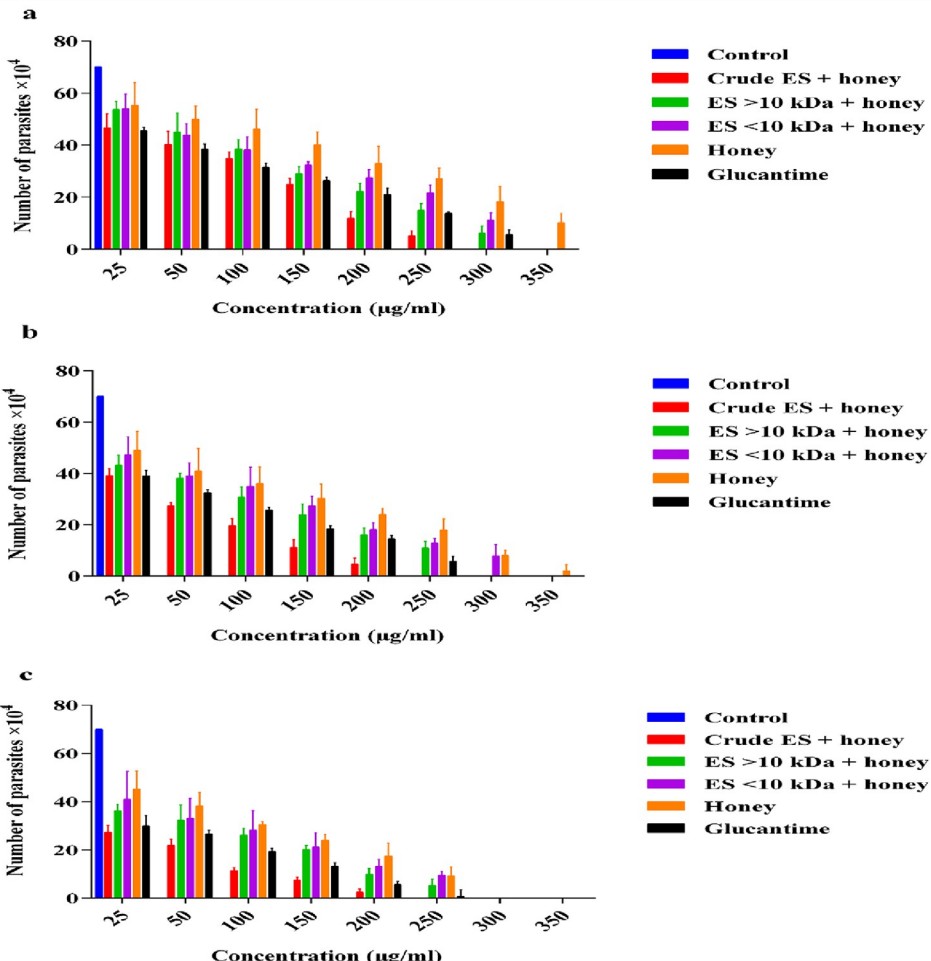

**Fig 3.** Mean (±SD) numbers of viable promastigotes exposed to various combinations of ES products at different doses and time intervals: (a) 24 h. (b) 48 h. (c) 72 h. Treatment groups significantly differed from the control group at $P < 0.05$.

## Discussion

Being a zoonotic diseases, the cutaneous leishmaniasis presents a major public health problem worldwide, especially in the tropics and subtropics [38, 39]. Due to high toxicity, cost, side effects and drug resistance incurred by pentavalent antimonials, researches are increasingly focused on finding cheaper drugs with minimal or no side effects [40, 41]. Insect-derived products including larval peptides and honey have been widely used to treat chronic wounds since ancient times [42]. The current study aimed to investigate leishmanicidal effects of *L. sericata* derived larval ES in combination with honey as a synergist on *L. major* promastigotes, the causative agent of cutaneous leishmaniasis (CL). In addition, the effects of two ES fractions together with honey were studied against the macrophage cell line J774A.1 and intracellular amastigotes.

Recent findings revealed that honey is a potent agent for cleaning infected wounds and has antimicrobial properties against a wide range of microbes, fungi and protozoa [33, 43, 44]. Its effectiveness on *Leishmania* parasite has been also reported [45]. In the present study, honey showed antileishmanial activity against *L. major* cells in accordance with others' findings [46].

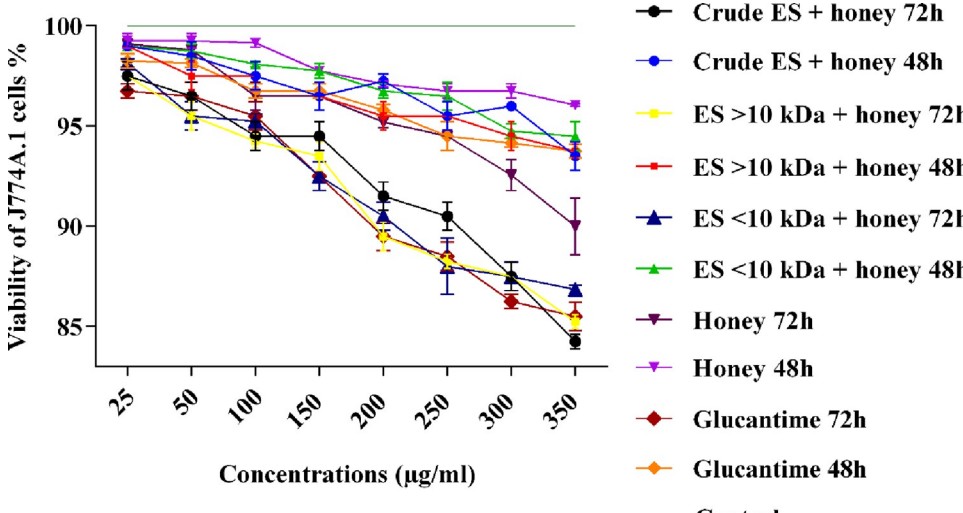

**Fig 4. Viability of macrophage cells exposed to different concentrations of larval products + honey as a synergist.**

Consistent with our study, honey incorporated with hydroalcoholic extract of *Nigella sativa* was found effective against *Leishmania* parasite [24]. Other studies reported the same results as in our study [28, 33, 47].

In agreement with previous reports, our results indicated the effects of larval ES plus honey on *L. major* promastigotes [37, 48, 49]. Comparison of $IC_{50}$ values of different combinations showed that larval ES plus honey exerted a higher toxic effect on promastigotes than each of the components as stand-alone treatment, though all treatments showed significant differences with the negative control (p < 0.0001). The same results were reported by other researchers [37, 49]. They even showed that highly concentrated larval ES exerted stronger lethal effects on exposed promastigotes than macrophage cells [37, 50].

A study involving cytotoxicity tests of larval ES products of *Calliphora vicinia* and *L. sericata* against macrophage (J774A.1) and *L. major* cells resulted in more than 40% toxicity to cells. This toxicity rate was higher than that reported in our study, although the ambiguity over the exact concentration used in aforementioned study makes the comparison difficult [51]. The report by Pinilla et al on the toxic effects of *Sarconesiopsis magellanica* larval ES on fibroblasts using the human lung-derived MRC5 cell line contradicted our reports. The authors showed that the larval ES was lethal to MRC5 cells at low concentrations [52]. They also

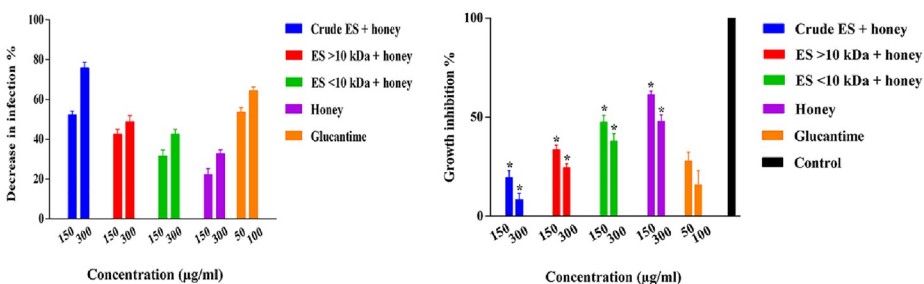

**Fig 5. Susceptibility of *L. major* amastigote grown in macrophages to *L. sericata* larval products plus honey as a synergist compared with glutamine at 72 h. post-treatment (level of significance * P < 0.05).**

**Table 1. Parameters indicating susceptibility of *L. major* amastigotes to *L. sericata* larval ES products with honey as a synergist.**

| Treatment | Dosages (µg/ml) | Infected cells % | Parasite load | Survival index |
|---|---|---|---|---|
| No treatment control | 0 | 85.37 ± 3.575 | 3.894 ± 0.091 | 273.97 ± 1.151 |
| Crude ES + honey | 150 | 40 ± 2.828 | 1.33 ± 0.087 | 53.45 ± 6.187 |
| | 300 | 20.67 ± 3.535 | 1.081 ± 0.155 | 22.09 ± 1.661 |
| ES >10 kDa + honey | 150 | 49 ± 2 | 1.868 ± 0.051 | 91.6 ± 6.233 |
| | 300 | 43.564 ± 2.516 | 1.535 ± 0.046 | 66.908 ± 8.11 |
| ES <10 kDa + honey | 150 | 58.34 ± 2.517 | 2.217 ± 0.06 | 129.426 ± 6.118 |
| | 300 | 49.01 ± 1.91 | 2.1 ± 0.014 | 103.04 ± 3.12 |
| Honey | 150 | 66.33 ± 2.515 | 2.517 ± 0.151 | 166.932 ± 4.323 |
| | 300 | 57.3 ± 1.527 | 2.273 ± 0.61 | 130.38 ± 3.994 |
| Glucantime | 50 | 39.66 ± 2.081 | 1.96 ± 0.038 | 77.89 ± 1.52 |
| | 100 | 30.23 ± 1.527 | 1.68 ± 0.061 | 51.957 ± 3.56 |

indicated that the survival rate of the cells did not change when exposed to larval ES products of *L. sericata* and *S. magellanica* at concentrations below 10 µg/ml, yet, the products became effective as the concentrations approached 20 µg/ml. This discrepancy with our results may be attributed to different insect strains, cell lines and laboratory conditions. However, the study by [37] who examined the effects of *L. sericata* larval products on peritoneal macrophages and J774A.1 cells, and that by [49] who examined ES products of *L. sericata* and *S. magellanica* on U937 cell line, were consistent with our results.

The larval products of *L. sericata* with honey had strong anti-*Leishmania* effects against intracellular amastigotes of *L. major*. All combinations reduced the infection rate and parasite load of infected macrophages, but the efficacy was even stronger when applying the crude ES plus honey and ES> 10 kDa plus honey compared with glucantime at 100 µg/mL. We found synergism between larval products and honey especially for the crude ES and ES> 10 kDa at sub-lethal concentrations against *L. major* infecting macrophage cell line J774A.1. The reduction in infection rate was more than 50% with the application of the crude ES plus honey and ES> 10 kDa plus honey at 150 µg/ml concentration and ES <10 kDa plus honey at 300 µg/ml concentration. However, these treatments were less toxic to the used macrophage cell line. The cell line maintained more than 88% viability when treated with the crude ES plus honey, ES> 10 kDa plus honey and ES< 10 kDa plus honey at 300 µg/ml concentrations.

Our finding was consistent with the report [37] in which the larval saliva and hemolymph of *L. sericata* larvae were tested on macrophage cell line J774A.1 infected with the *L. tropica* parasite. A similar result was obtained by examination of larval ES of *L. sericata* and *S. magellanica* on U937 cell line infected with *Leishmania panamensis* parasite [49]. The larval ES produced sharper decrease in parasite survival index at higher concentrations. This was consistent with the study by Rahimi et al [37], but contrasted earlier studies in which susceptibility of J774A.1. cell line infected with *L. major* amastigotes and U937 human cell line infected with *L. panamensis* amastigotes was evaluated [49, 51].

The parasite load and survival index values of macrophage cells infected with *L. major* were lower upon treatment with the crude ES plus honey and ES> 10 kDa plus honey compared with ES< 10 kDa plus honey or honey alone. Furthermore, significant decreases in parasite loads and survival indexes were observed in treated groups compared to the untreated control ($P = 0.05$), which was consistent with findings of Rahimi et al [37]. Our study confirmed the result of Sherafati et al that honey acts as a synergist with ES products [30]. Indeed, leishmanicidal effects of larval ES products in combination with honey to intracellular amastigotes provide a promising basis for *in vivo* experimentation in future.

A number of studies have shown anti-leishmanial effects of larval products on *Leishmania* species such as *L. amazonensis* [53], *L. tropica* [48], *L. major* [54], and *L. panamensis* [29] under in vitro and in vivo conditions. Our study provides a new evidence on leishmaniacidal and potential therapeutic effects of larval ES products of *L. sericata* combined with honey as a synergist to *L. major* promastigotes and intracellular amastigotes.

## Conclusions

To our knowledge, this is the first study dealing with the effects of *L. sericata* larval-ES products in combination with *A. mellifera* honey on different infective stages of *L. major*. The study followed a quantitative approach to test the cytotoxicity of ES products plus honey on mice macrophage cell line J774A.1. It should be noted that the crude ES plus honey at a concentration of 300 μg/ml was more effective than glutamine as a standard drug at a concentration of 100 μg/ml. Also, there was no significant difference between toxicity of ES> 10 kDa plus honey with that of glucantime. Our results showed that there was a synergistic effect between honey and larval ES products for the inhibition of intracellular and extracellular cells of *L. major*. The larval excretion/excretion product contains a mixture of proteolytic, glycolytic, lipolytic, as well as nucleoid enzymes which are released on wound surfaces for maggot therapy to take place using *L. sericata* larvae. The following activation of collagenases, gelatinases, sterolisins, MMPs (MMP-2 and MMP-9) enables many biological and pathological processes to accelerate wound healing [55]. However, the inhibitory action of ES components of *Lucilia sericata* in combination with honey or its derivatives such as royal jelly or wax in against *Leishmania major* needs further investigation both under in vitro and in vivo conditions.

## Supporting information

**S1 File.**
(PDF)

## Acknowledgments

The authors would like to thank the Department of Parasitology and Medical Entomology and the Student Research Committee of Tarbiat Modares University for their assistance in this project.

## Author Contributions

**Conceptualization:** Jila Sherafati, Mohammad Saaid Dayer, Fatemeh Ghaffarifar.

**Data curation:** Jila Sherafati, Fatemeh Ghaffarifar.

**Formal analysis:** Mohammad Saaid Dayer, Kamran Akbarzadeh, Majid Pirestani.

**Funding acquisition:** Mohammad Saaid Dayer.

**Investigation:** Jila Sherafati, Fatemeh Ghaffarifar.

**Methodology:** Mohammad Saaid Dayer, Fatemeh Ghaffarifar, Kamran Akbarzadeh, Majid Pirestani.

**Project administration:** Mohammad Saaid Dayer.

**Resources:** Kamran Akbarzadeh, Majid Pirestani.

**Software:** Jila Sherafati, Majid Pirestani.

**Supervision:** Mohammad Saaid Dayer.

**Validation:** Mohammad Saaid Dayer, Fatemeh Ghaffarifar, Kamran Akbarzadeh, Majid Pirestani.

**Visualization:** Fatemeh Ghaffarifar, Kamran Akbarzadeh, Majid Pirestani.

**Writing – original draft:** Jila Sherafati.

**Writing – review & editing:** Mohammad Saaid Dayer, Fatemeh Ghaffarifar, Kamran Akbarzadeh, Majid Pirestani.

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
