## [Decision Letter · Decision Letter 0]

21 Sep 2022

PONE-D-22-17517Evaluating leishmanicidal activity of Lucilia sericata larval products in combination with Apis mellifera honey using an in vitro modelPLOS ONE

Dear Dr. Dayer,

Thank you for submitting your manuscript to PLOS ONE. After careful consideration, we feel that it has merit but does not fully meet PLOS ONE’s publication criteria as it currently stands. Therefore, we invite you to submit a revised version of the manuscript that addresses the points raised during the review process.

We look forward to receiving your revised manuscript.

Kind regards,

Alireza Badirzadeh

Academic Editor

PLOS ONE

Journal Requirements:

“MSD has received a grant (Grant no: Med/8288) from Tarbiat Modares University, Tehran, Iran

https://www.modares.ac.ir/en

No”

“The authors would like to thank all staff of the Department of Parasitology and Medical Entomology and the Student Research Committee as well as the Faculty of Medical Sciences of Tarbiat Modares University for their support.”

 “MSD has received a grant (Grant no: Med/8288) from Tarbiat Modares University, Tehran, Iran

https://www.modares.ac.ir/en

No”

6. Please amend either the title on the online submission form (via Edit Submission) or the title in the manuscript so that they are identical.

7. We note you have included a table to which you do not refer in the text of your manuscript. Please ensure that you refer to Table 2 in your text; if accepted, production will need this reference to link the reader to the Table.

8. Please include a copy of Table 1 which you refer to in your text on page 11.

Reviewers' comments:

Reviewer's Responses to Questions

**Comments to the Author**

1. Is the manuscript technically sound, and do the data support the conclusions?

Reviewer #1: Yes

Reviewer #2: Yes

2. Has the statistical analysis been performed appropriately and rigorously? 

Reviewer #1: Yes

Reviewer #2: Yes

3. Have the authors made all data underlying the findings in their manuscript fully available?

Reviewer #1: Yes

Reviewer #2: Yes

4. Is the manuscript presented in an intelligible fashion and written in standard English?

Reviewer #1: Yes

Reviewer #2: No

5. Review Comments to the Author

Reviewer #1: Dear authors;

First of all, I congratulate you on carrying out this valuable study. But in my opinion, some improvements are necessary to enrich the manuscript. So, I have listed my comments.

1. The title should be more impressive. Therefore, I suggest “Leishmaniacidal potential of Lucilia sericata products and Apis mellifera honey combination: an in vitro evaluation.”

2. Language editing in the text of the manuscript is required.

3. Please employ MeSh to check your keywords; I think J744 is not an appropriate keyword.

4. In the introduction, you mentioned the existence of three forms of the disease, but it is better to say that there are three main forms of the disease.

5. According to your manuscript, new cases of CL have been estimated at 1.2 million. Still, the website of WHO says,” It is estimated that between 600 000 to 1 million new cases occur worldwide annually”.

6. Although in lines 210 and 216, you indicate the treatment of honey plus Glucantime, I could not find more information in “Material and Method” and table 1. Please clarify the treatment.

7. Despite only one table in the manuscript, you named it “Table 2”! Please correct it.

8. You have mentioned further studies are needed to detect the active components of larval ES and honey. Please mention probable active ingredients, and discuss them in brief.

Best wishes

Reviewer #2: The manuscript entitled " Evaluating leishmanicidal activity of Lucilia sericata larval products in combination with Apis mellifera honey using an in vitro model" describes the effects of larval products of L. sericata mixed with honey on Leishmania major promastigotes and infected macrophages. The following contains some point that should be addressed by the authors.

The Leishmania species (major) used in this study should be included in the title.

The authors should describe the experimental processes to check the purity of the

two fractions of ES (>10 and <10 kDa).

Figure is not necessary.

The viable amastigotes were not evaluated, only % of infected macrophages and number of intracellular amastigotes in J774 macrophage cultures.

A screening for lipopolysaccharide (LPS/endotoxin) contamination in larval products and honey should be performed

Images of treated and untreated promastigotes and infected macrophages could be included in the manuscript.

The authors suggested further studies to detect the active components of larval ES and honey as well as the mechanism of inhibition on the Leishmania parasite (line 314). Some suggestion that these issues could be discussed in the manuscript.

6. PLOS authors have the option to publish the peer review history of their article (what does this mean?). If published, this will include your full peer review and any attached files.

Reviewer #1: No

Reviewer #2: No

---

## [Author Response · Author response to Decision Letter 0]

21 Dec 2022

Journal Requirements:

Our response: 

We did the required modification according to PLOS ONE style patterns. Thanks.

2. Please provide additional details regarding participant consent. In the ethics statement in the Methods and online submission information, please ensure that you have specified (1) whether consent was informed and (2) what type you obtained (for instance, written or verbal, and if verbal, how it was documented and witnessed). If your study included minors, state whether you obtained consent from parents or guardians. If the need for consent was waived by the ethics committee, please include this information. If you are reporting a retrospective study of medical records or archived samples, please ensure that you have discussed whether all data were fully anonymized before you accessed them and/or whether the IRB or ethics committee waived the requirement for informed consent. If patients provided informed written consent to have data from their medical records used in research, please include this information.

Our response:

Our study did not involve human or animal specimens. However, the undertaken methodologies of the study were approved by the Ethics Committee of Tarbiat Modares University. 

 “MSD has received a grant (Grant no: Med/8288) from Tarbiat Modares University, Tehran, Iran https://www.modares.ac.ir/en No” Please state what role the funders took in the study. If the funders had no role, please state: "The funders had no role in study design, data collection and analysis, decision to publish, or preparation of the manuscript."

Our response: 

Thanks for your suggestion. The sentence was edited, and included in the cover letter as instructed. 

“The authors would like to thank all staff of the Department of Parasitology and Medical Entomology and the Student Research Committee as well as the Faculty of Medical Sciences of Tarbiat Modares University for their support.”

“MSD has received a grant (Grant no: Med/8288) from Tarbiat Modares University, Tehran, Iran https://www.modares.ac.ir/en

 No” Please include your amended statements within your cover letter; we will change the online submission form on your behalf.

Our response: 

Thanks for your assistance. The statement was included in the cover letter as required.

Upon re-submitting your revised manuscript, please upload your study’s minimal underlying data set as either Supporting Information files or to a stable, public repository and include the relevant URLs, DOIs, or accession numbers within your revised cover letter. For a list of acceptable repositories, please see http://journals.plos.org/plosone/s/data-availability#loc recommended-repositories. Any potentially identifying patient information must be fully anonymized.

Important: If there are ethical or legal restrictions to sharing your data publicly, please explain these restrictions in detail. Please see our guidelines for more information on what we consider unacceptable restrictions to publicly sharing data: http://journals.plos.org/plosone/s/data-availability#loc unacceptable-data-access-restrictions. Note that it is not acceptable for the authors to be the sole named individuals responsible for ensuring data access. We will update your Data Availability statement to reflect the information you provide in your cover letter.

Our response: 

The supporting data was provided as a separate file and uploaded. 

6. Please amend either the title on the online submission form (via Edit Submission) or the title in the manuscript so that they are identical.

Our response:

Sure. We have done so. 

7. We note you have included a table to which you do not refer in the text of your manuscript. Please ensure that you refer to Table 2 in your text; if accepted, production will need this reference to link the reader to the Table.

Our response:

Thanks for your attention. We included a reference for the sole table in the main text as required. 

8. Please include a copy of Table 1 which you refer to in your text on page 11.

Our response: 

Sure. We did so. 

Reviewers' comments:

Reviewer's Responses to Questions

Comments to the Author

1. Is the manuscript technically sound, and do the data support the conclusions?

Reviewer #1: Yes

Reviewer #2: Yes

2. Has the statistical analysis been performed appropriately and rigorously?

Reviewer #1: Yes

Reviewer #2: Yes

3. Have the authors made all data underlying the findings in their manuscript fully available?

The PLOS Data policy [1] requires authors to make all data underlying the findings described in their manuscript fully available without restriction, with rare exception (please refer to the Data Availability Statement in the manuscript PDF file). The data should be provided as part of the manuscript or its supporting information, or deposited to a public repository. For example, in addition to summary statistics, the data points behind means, medians and variance measures should be available. If there are restrictions on publicly sharing data—e.g. participant privacy or use of data from a third party—those must be specified.

Reviewer #1: Yes

Reviewer #2: Yes

4. Is the manuscript presented in an intelligible fashion and written in standard English?

Reviewer #1: Yes

Reviewer #2: No

5. Review Comments to the Author

Reviewer #1: Dear authors;

First of all, I congratulate you on carrying out this valuable study. But in my opinion, some improvements are necessary to enrich the manuscript. So, I have listed my comments.

1. The title should be more impressive. Therefore, I suggest “Leishmaniacidal potential of Lucilia sericata products and Apis mellifera honey combination: an in vitro evaluation.”

Our response:

Thanks. We agree. The title was changed as kindly suggested. 

2. Language editing in the text of the manuscript is required.

Our response:

The whole manuscript has now been thoroughly revised for English style. Thank you. 

3. Please employ MeSh to check your keywords; I think J744 is not an appropriate keyword.

Our response:

Thank you for your attention. We substituted it with the word “macrophage”.

4. In the introduction, you mentioned the existence of three forms of the disease, but it is better to say that there are three main forms of the disease.

Our response:

Thank you. Corrected as kindly suggested.

5. According to your manuscript, new cases of CL have been estimated at 1.2 million. Still, the website of WHO says,” It is estimated that between 600 000 to 1 million new cases occur worldwide annually”.

Our response:

Thanks for your attention. The correction was made and a new reference added.

6. Although in lines 210 and 216, you indicate the treatment of honey plus Glucantime, I could not find more information in “Material and Method” and table 1. Please clarify the treatment.

Our response:

Thanks for your attention. We are sorry for this mistake. In fact, we did not have honey + glucantime treatment, but used glucantime as the first line treatment to compare with. So, we have corrected this mistake in the main text. 

7. Despite only one table in the manuscript, you named it “Table 2”! Please correct it.

Our response:

Thanks. It has been corrected.

8. You have mentioned further studies are needed to detect the active components of larval ES and honey. Please mention probable active ingredients, and discuss them in brief.

Best wishes

Our response: 

Thanks. We have briefly discussed it the text. 

Reviewer #2: 

The manuscript entitled " Evaluating leishmanicidal activity of Lucilia sericata larval products in combination with Apis mellifera honey using an in vitro model" describes the effects of larval products of L. sericata mixed with honey on Leishmania major promastigotes and infected macrophages. The following contains some point that should be addressed by the authors.

The Leishmania species (major) used in this study should be included in the title.

Our response: 

Thanks for your suggestion. We added the species name in the title.

The authors should describe the experimental processes to check the purity of the two fractions of ES (>10 and <10 kDa). Figure is not necessary.

Our response:

As kindly suggested, we have added a section to describe the evaluation method for protein band purity in the main text.

The viable amastigotes were not evaluated, only % of infected macrophages and number of intracellular amastigotes in J774 macrophage cultures.

Our response:

Yes. We did not check the amastigote viability as correctly noticed. But we checked the percentage of growth inhibition, and the infection rate which presented in both Table 1 and Figure 5. So, the necessary corrections have now been introduced.

 A screening for lipopolysaccharide (LPS/endotoxin) contamination in larval products and honey should be performed Images of treated and untreated promastigotes and infected macrophages could be included in the manuscript.

Our response:

Thanks for the suggestion. As the ES products and honey, used in this study, were freshly prepared and immediately applied to perform the experimentations, we did not attempt to test LPS contamination. In fact, such tests had not been undertaken by many references which were examined for this paper such as no. 37 and 49. In addition the recent profiling of ES proteins and peptides of Lucilia sericata did not report any LPS trace as an integral component, rather they proved the suppressive action of ES to bacterial LPS. The same was reported by many recent researches for honey being protective against LPS. However, as the respected reviewer has rightly mentioned, the LPS role as an exogenous contaminant may not be ruled out. This is particularly important when any ES products are to be developed as prescription drugs. We will surely consider the suggestion in our future researches. Many thanks. 

The authors suggested further studies to detect the active components of larval ES and honey as well as the mechanism of inhibition on the Leishmania parasite (line 314). Some suggestion that these issues could be discussed in the manuscript.

Our response: 

Many thanks. Some explanations were added in the discussion section. 

6. PLOS authors have the option to publish the peer review history of their article (what does this mean? [2]). If published, this will include your full peer review and any attached files.

Do you want your identity to be public for this peer review? For information about this choice, including consent withdrawal, please see our Privacy Policy [3].

Reviewer #1: No

Reviewer #2: No

---

## [Decision Letter · Decision Letter 1]

30 Jan 2023

PONE-D-22-17517R1Evaluating leishmanicidal effects of Lucilia sericata products in combination with Apis mellifera honey using an in vitro modelPLOS ONE

Dear Dr. Saaid Dayer,

Thank you for submitting your manuscript to PLOS ONE. After careful consideration, we feel that it has merit but does not fully meet PLOS ONE’s publication criteria as it currently stands. Therefore, we invite you to submit a revised version of the manuscript that addresses the points raised during the review process.

We look forward to receiving your revised manuscript.

Kind regards,

Alireza Badirzadeh

Academic Editor

PLOS ONE

Journal Requirements:

Reviewers' comments:

Reviewer's Responses to Questions

**Comments to the Author**

1. If the authors have adequately addressed your comments raised in a previous round of review and you feel that this manuscript is now acceptable for publication, you may indicate that here to bypass the “Comments to the Author” section, enter your conflict of interest statement in the “Confidential to Editor” section, and submit your "Accept" recommendation.

Reviewer #1: All comments have been addressed

Reviewer #2: All comments have been addressed

2. Is the manuscript technically sound, and do the data support the conclusions?

Reviewer #1: Yes

Reviewer #2: Yes

3. Has the statistical analysis been performed appropriately and rigorously? 

Reviewer #1: Yes

Reviewer #2: Yes

4. Have the authors made all data underlying the findings in their manuscript fully available?

Reviewer #1: Yes

Reviewer #2: Yes

5. Is the manuscript presented in an intelligible fashion and written in standard English?

Reviewer #1: Yes

Reviewer #2: Yes

6. Review Comments to the Author

Reviewer #1: Dear authors;

First, I congratulate you due to carrying out this valuable rearch, and thank you to apply all comments.

Sincerely yours

Reviewer #2: There is no satisfactory section to describe the evaluation method for protein band purity in the revised text.

In addition, figure 1 show only a bovine serum albumin standard curve not a concentration of L. sericata larval products curve. The authors describe in the text: the average net absorbance at 595 nm for seven standard dilutions from low to high…. Average protein concentrations for the crude ES, ES >10 kDa, and ES <10 kDa were 809.8, 680.6, and 385.3 µg/ml …(Figure 1) (lines 175-180).

Please rewrite this section and consider do not show a BSA standard curve to describe protein concentrations of larval products. A table or just the description in the text are good options.

7. PLOS authors have the option to publish the peer review history of their article (what does this mean?). If published, this will include your full peer review and any attached files.

Reviewer #1: No

Reviewer #2: No

---

## [Author Response · Author response to Decision Letter 1]

21 Feb 2023

Reviewers' comments:

Reviewer's Responses to Questions

Comments to the Author

1. If the authors have adequately addressed your comments raised in a previous round of review and you feel that this manuscript is now acceptable for publication, you may indicate that here to bypass the “Comments to the Author” section, enter your conflict of interest statement in the “Confidential to Editor” section, and submit your "Accept" recommendation.

Reviewer #1: All comments have been addressed

Reviewer #2: All comments have been addressed

2. Is the manuscript technically sound, and do the data support the conclusions?

Reviewer #1: Yes

Reviewer #2: Yes

3. Has the statistical analysis been performed appropriately and rigorously?

Reviewer #1: Yes

Reviewer #2: Yes

4. Have the authors made all data underlying the findings in their manuscript fully available?

The PLOS Data policy [1] requires authors to make all data underlying the findings described in their manuscript fully available without restriction, with rare exception (please refer to the Data Availability Statement in the manuscript PDF file). The data should be provided as part of the manuscript or its supporting information, or deposited to a public repository. For example, in addition to summary statistics, the data points behind means, medians and variance measures should be available. If there are restrictions on publicly sharing data—e.g. participant privacy or use of data from a third party—those must be specified.

Reviewer #1: Yes

Reviewer #2: Yes

5. Is the manuscript presented in an intelligible fashion and written in standard English?

Reviewer #1: Yes

Reviewer #2: Yes

6. Review Comments to the Author

Reviewer #1: Dear authors;

First, I congratulate you due to carrying out this valuable rearch, and thank you to apply all comments.

Sincerely yours

Our response: 

Thanks you so much.

Reviewer #2: There is no satisfactory section to describe the evaluation method for protein band purity in the revised text.

Our response: 

Thank you for the comment. We have added a section describing the evaluation method of protein purity as we did in our earlier study.

In addition, figure 1 show only a bovine serum albumin standard curve not a concentration of L. sericata larval products curve. The authors describe in the text: the average net absorbance at 595 nm for seven standard dilutions from low to high…. Average protein concentrations for the crude ES, ES >10 kDa, and ES <10 kDa were 809.8, 680.6, and 385.3 µg/ml … (Figure 1) (lines 175-180). 

Please rewrite this section and consider do not show a BSA standard curve to describe protein concentrations of larval products. A table or just the description in the text are good options.

Our response: 

Thank you for your attention

We have now introduced the required changes in Fig. 1. The previous graph was drawn using online site (at https://www.aatbio.com), which unfortunately did not have the ability to accommodate the results of larval products. Therefore. we have used Prism software, Saturation Binding Data Graph section, to combine and redraw the corresponding data. As shown, the graph now indicates the standard curve as well as the OD’s obtained from crude and fractionated larval extracts.The data analysis part of this point was also added.

---

## [Decision Letter · Decision Letter 2]

7 Mar 2023

Evaluating leishmanicidal effects of Lucilia sericata products in combination with Apis mellifera honey using an in vitro model

PONE-D-22-17517R2

Dear Dr. Mohammad Saaid Dayer,

We’re pleased to inform you that your manuscript has been judged scientifically suitable for publication and will be formally accepted for publication once it meets all outstanding technical requirements.

Kind regards,

Alireza Badirzadeh

Academic Editor

PLOS ONE

Additional Editor Comments (optional):

Reviewers' comments:

Reviewer's Responses to Questions

**Comments to the Author**

1. If the authors have adequately addressed your comments raised in a previous round of review and you feel that this manuscript is now acceptable for publication, you may indicate that here to bypass the “Comments to the Author” section, enter your conflict of interest statement in the “Confidential to Editor” section, and submit your "Accept" recommendation.

Reviewer #1: All comments have been addressed

Reviewer #2: All comments have been addressed

2. Is the manuscript technically sound, and do the data support the conclusions?

Reviewer #1: Yes

Reviewer #2: Yes

3. Has the statistical analysis been performed appropriately and rigorously? 

Reviewer #1: Yes

Reviewer #2: Yes

4. Have the authors made all data underlying the findings in their manuscript fully available?

Reviewer #1: Yes

Reviewer #2: Yes

5. Is the manuscript presented in an intelligible fashion and written in standard English?

Reviewer #1: Yes

Reviewer #2: Yes

6. Review Comments to the Author

Reviewer #1: Dear authors;

I believe we must try to develop natural medicines, so your valuable manuscript could get my interest, and you adopted all of my primary comments. Therefore, the current version is completely eligible to be published in the journal.

Sincerely yours

Reviewer #2: (No Response)

7. PLOS authors have the option to publish the peer review history of their article (what does this mean?). If published, this will include your full peer review and any attached files.

Reviewer #1: No

Reviewer #2: No

---

## [Editor Report · Acceptance letter]

15 Mar 2023

PONE-D-22-17517R2 

Evaluating leishmanicidal effects of *Lucilia sericata* products in combination with *Apis mellifera* honey using an in vitro model 

Dear Dr. Dayer:

I'm pleased to inform you that your manuscript has been deemed suitable for publication in PLOS ONE. Congratulations! Your manuscript is now with our production department. 

Kind regards, 

on behalf of

Dr. Alireza Badirzadeh 

Academic Editor

PLOS ONE